# Positive and negative emotions during the COVID-19 pandemic: A longitudinal survey study of the UK population

Lan Li[1]*, Ava Sullivan[1,2], Anwar Musah[1,3], Katerina Stavrianaki[1,4], Caroline E. Wood[1], Philip Baker[1,5], Patty Kostkova[1]

1 Centre for Digital Public Health in Emergencies, Institute for Risk and Disaster Reduction, University College London, London, United Kingdom, 2 EcoHealth Alliance, New York, New York, United States of America, 3 Department of Geography, University College London, London, United Kingdom, 4 Department of Statistical Science, University College London, London, United Kingdom, 5 Crisis Response, British Red Cross, London, United Kingdom

* lan.li.19@ucl.ac.uk

**Data Availability Statement:** There are ethical restrictions related to participant confidentiality which prevent the public sharing of minimal data for this study. However, the minimal data are

## Abstract

The COVID-19 pandemic has had a profound impact on society; it changed the way we work, learn, socialise, and move throughout the world. In the United Kingdom, policies such as business closures, travel restrictions, and social distance mandates were implemented to slow the spread of COVID-19 and implemented and relaxed intermittently throughout the response period. While negative emotions and feelings such as distress and anxiety during this time of crisis were to be expected, we also see the signs of human resilience, including positive feelings like determination, pride, and strength. A longitudinal study using online survey tools was conducted to assess people's changing moods during the pandemic in the UK. The Positive and Negative Affect Schedule (PANAS) was used to measure self-reported feelings and emotions throughout six periods (phases) of the study from March 2020 to July 2021. A total of 4,222 respondents participated in the survey, while a sub-group participated in each of the six study phases (n = 167). The results were analysed using a cross-sectional study design for the full group across each study phase, while prospective cohort analysis was used to assess the subset of participants who voluntarily answered the survey in each of the six study phases (n = 167). Gender, age and employment status were found to be most significant to PANAS score, with older people, retirees, and women generally reporting more positive moods, while young people and unemployed people generally reported lower positive scores and higher negative scores, indicating more negative emotions. Additionally, it was found that people generally reported higher positive feelings in the summer of 2021, which may be related to the relaxation of COVID-19-related policies in the UK as well as the introduction of vaccines for the general population. This study is an important investigation into what allows for positivity during a crisis and gives insights into periods or groups that may be vulnerable to increased negative states of emotions and feelings.

available upon request from University College London (UCL) Institute of Risk and Disaster Reduction (IRDR) Centre for Digital Public Health in Emergencies (dPHE) via email (irdr.dphe@ucl.ac.uk), or through their website (https://www.ucl.ac.uk/risk-disaster-reduction/ucl-irdr-centre-digital-public-health-emergencies-dphe), for researchers who meet the criteria for access to confidential data.

**Funding:** LL was partially supported by China Scholarship Council (File No. 202008060009). the funders had no role in study design, data collection and analysis, decision to publish, or preparation of the manuscript.

**Competing interests:** The authors have declared that no competing interests exist.

## Introduction

Natural disasters have been posited as substantial sources of life stress for both individuals and communities [1]. When confronted with the eminent stressor of disaster or emergencies, individuals commonly experience a range of emotions, including fear, anxiety, sadness, and anger [2]. The COVID-19 pandemic, which emerged in the UK in January 2020, has exposed Britons to prolonged stress related to the ongoing health crisis [1]. Various studies have provided evidence that such catastrophic events can result in significant emotional distress and have long-lasting adverse effects on an individual's subjective well-being. Understanding emotional well-being in the context of disasters becomes crucial in assessing individuals' coping abilities during emergencies and determining whether their well-being is compromised to the extent that it impairs their normal functioning. In addition, mental well-being is considered a critical element of community resilience during the response and recovery phase of a disaster [3].

Between March 2020 and July 2021, the UK government issued a series of policies and restrictions to curb the spread of the virus, such as social distancing, lockdowns, and travel bans [4–7]. With these back-and-forth policy changes, people's mental stress and emotions have risen and fallen, leading to anxiety, depression, and negative emotions [8, 9]. During this time, people have experienced myriad feelings related to the pandemic, including loneliness from lockdown, fear of illness, and uncertainty due to changing rules and regulations [10]. However, given the prolonged period over which the COVID-19 pandemic occurred and the sheer resilience of humans during disasters, there have also been moments of joy, inspiration, and gratitude. Surprisingly, these moments are not simply despite the disaster, but rather positive moments directly related to the changes and response to the pandemic [11].

Many recent studies using cross-sectional surveys examined the emotional impact of the COVID-19 pandemic among children [12, 13], college students [14], healthcare workers [15, 16], and the general population [8]. It is widely acknowledged that the pandemic significantly influences people's mental health due to the health crisis and uncertainties, and these effects have uneven effects on certain vulnerable groups [9]. Longitudinal cohort studies carried out in the UK demonstrated that mental health deteriorated in the early stage of lockdown (31 March to 9 April 2020) [17, 18]. Other longitudinal evidence gathered from Germany [19], China [20, 21], Austria [22], and France [23] and systematic reviews [24–26] demonstrated that the impact of lockdown on emotions varies among countries and timings. However, since most of the longitudinal studies focused on the initial stage of the pandemic and fewer tracked the post-pandemic, it is hard to investigate the relationship between the policy change over the whole period and how individuals' feelings changed during different periods of the prolonged pandemic. This element may be particularly important to understand the broader public's tolerance and well-being during a disaster and to understand public health and mental health trade-offs inherent in disaster response efforts [27].

Throughout the COVID-19 pandemic, policies and public health measures have been enacted to curb the spread of the virus in communities. These policies often dictated the type and frequency of non-essential activities and movement, such as travel bans, group size limits, and venue closures. In general, the severity and scale of the policies themselves are related to the severity of COVID-19 in the country. As the pandemic spread throughout Britain, public health measures rushed to respond to the increased morbidity and mortality within the UK. This study aims to assess the change in emotions and feelings during the COVID-19 pandemic to the restrictions that were in place during the number of phases of the pandemic throughout the study period. During the study period, the UK underwent a series of 'lockdowns' representing the most stringent measures of restricted movement, business closures, and gathering limitations. During the same period, Britons also experienced periods of more relaxed

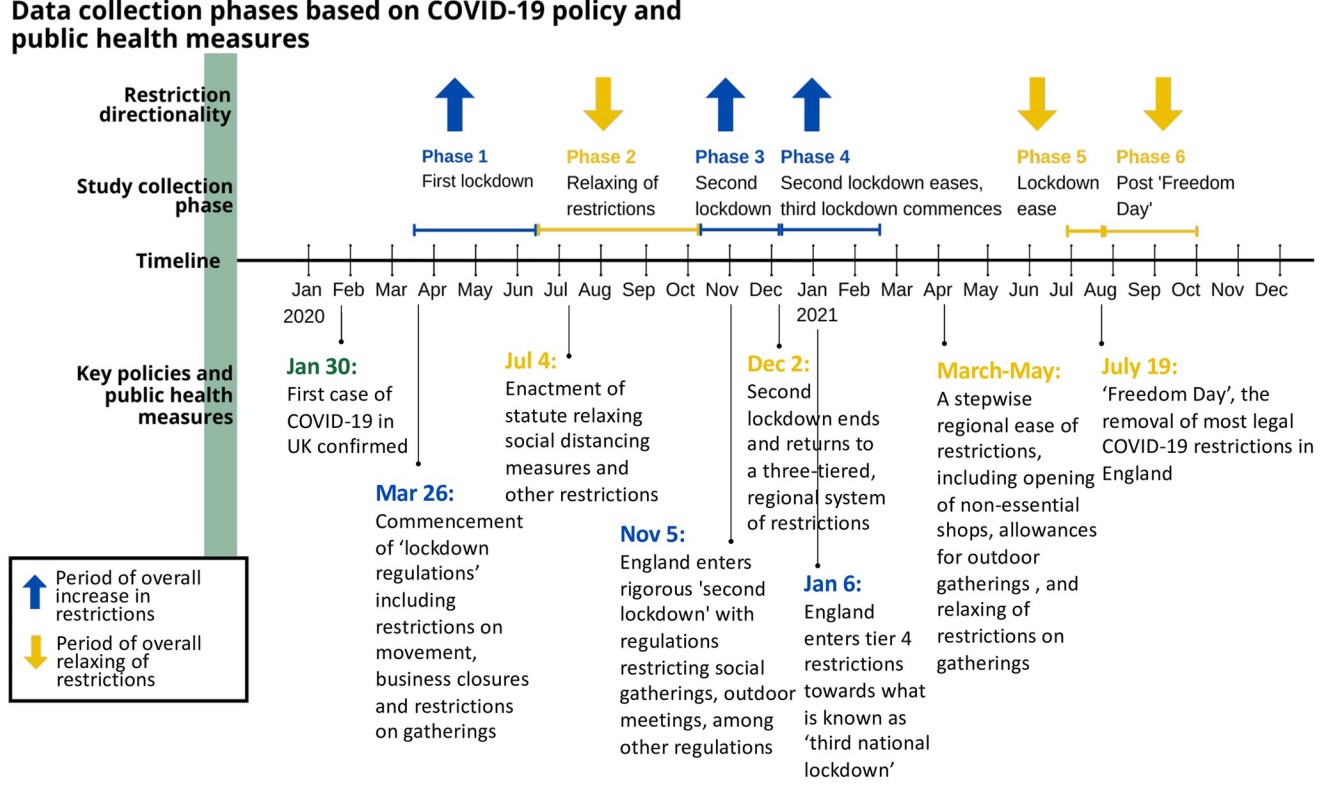

**Fig 1. Data collection phases based on COVID-19 policy and public health measures.**

restrictions, when risk levels were deemed in decline, and social freedoms were restored to certain extents based on national or local risks. These periods of fluctuating restrictions and reversals are mapped in Fig 1. The longitudinal surveys were collected at six study points, called 'Phases', throughout this period of fluctuating public health restrictions. During each phase, the predominant qualities of the time related to COVID-19 restrictions allowed researchers to understand a potential driver for positive or negative feelings measured by the standard Positive and Negative Affect Schedule-Trait scale (PANAS).

This study aims to measure the emotional changes experienced throughout the pandemic longitudinally across people living in the UK. The study period reflects various COVID-19 caseloads in the UK and various policies set forth to curve the spread.

## Methodology

### Participants and study design

This study is a part of the 'To Zoom or Not to Zoom' project, which is designed to investigate the influence of the COVID-19 pandemic on lifestyle, activity changes, and related emotions throughout the UK population. While the analyses of changing activities are discussed in another paper [28], this analysis is focused on the emotional response to changing public health restrictions during the COVID-19 pandemic.

The project consists of six surveys conducted fully online, covering the course of the pandemic and various stages of lockdown over fifteen months (April 2020 to July 2021). The sampling was opportunistic, drawn from UK residents through Facebook advertising, mutual aid

groups, and other social media channels. Participants were required to be 18 years of age or older. The convenience sampling method was used to maximise the sample size within the limited timeframe and resources. In April 2020, 3240 participants voluntarily completed the survey and provided their email addresses for follow-up. The five follow-up surveys were carried out through sending emails to the same email lists in May 2020 (n = 1399), October 2020 (n = 856), December 2020 (n = 1050), June 2021 (n = 1298) and July (n = 1036). New participants were recruited through Facebook advertising during the third survey period in October 2020 (n = 1762) and the fourth survey in December 2020 (n = 143). All the survey data was collected via SurveyMonkey.com.

The participants were asked to respond to demographic questions in each survey, including age, gender, education, employment, demographics, and COVID-19 infection history. The survey also includes questions regarding the frequency and mode of access for 16 different activities, analysed separately to assess the behaviour change during the pandemic [29]. The completion of these questions was optional. The full questionnaire is enclosed in the S1 File.

## PANAS scale

The standard Positive and Negative Affect Schedule-Trait (PANAS [28]) scale was used to measure mood and feelings. The PANAS scale is a self-reported psychological scale with ten positive markers and ten negative markers [30]. The Positive Attribute (PA) subscale reflects the extent to which a person feels interested, excited, strong, enthusiastic, proud, alert, inspired, determined, attentive, and active. The Negative Attribute (NA) subscale includes stress, upset, guilty, scared, hostile, irritable, ashamed, nervous, jittery, and afraid. All the items are rated on a scale ranging from 1 ('very slightly or not at all') to 5 ('extremely'). It is the most widely and frequently used scale to assess positive and negative affect and has shown excellent psychometric properties in the general population [31–33].

## Data cleaning and analysis

To analyse the data, the responses were divided into six study phases according to the timeline of the relevant public health policies in England, as shown in Fig 1. The records that spanned the node f phases were retroactively categorised based on the collection date. All responses with duplicate email addresses were assessed in each phase, and the most completed records were kept. Next, only the records that responded to all 20 items were included. After the deduplication and removal of the incomplete records, there were 4,222 unique participants included in the cross-sectional study and 167 participants agreed to participate in every stage of the project, completing all six surveys included in the prospective cohort study. Table 1

**Table 1. Details of data cleaning in each phase.**

| Study Phase | | Phase 1 | Phase 2 | Phase 3 | Phase 4 | Phase 5 | Phase 6 |
|---|---|---|---|---|---|---|---|
| **Data collected** | No. of responses from 1st recruitment | N = 3240 | N = 1399 | N = 856 | N = 1050 | N = 1298 | N = 1036 |
| | No. of responses from 2nd recruitment | | | N = 1762 | N = 143 | | |
| | Total responses collected | N = 3240 | N = 1399 | N = 2618 | N = 1193 | N = 1298 | N = 1036 |
| **No. of responses filtered out** due to incomplete records, duplicated email address identifier, low data quality (i.e.. response time<100s) | | N = 553 | N = 1056[a] | N = 664 | N = 134 | N = 158 | N = 146 |
| **Final included sample for analysis** | Sample size | N = 2687 | N = 343 | N = 1954 | N = 1059 | N = 1140 | N = 890 |
| | **Total responses** | N = 8073 | | | | | |
| | **No. of unique participants** | N = 4222 | | | | | |

[a]: The large filter-out rate in Phase 2 due to the majority of the data spanning the node of Phase 1 and Phase 2 and being repeated with Phase 1 participants.

presents the sample size and data cleaning details for each phase, including the final number of participants included in the analysis.

Two different study designs have been used to interpret and analyse the survey data. First, a cross-sectional study design was used to understand the general emotional affect and parse the results based on demographic factors throughout the six phases. Second, a prospective cohort study design was used to explore how individuals adapted to the pandemic throughout the phases among a subset of repeated respondents. The latter study design allowed researchers to observe the changes in emotional state over time amongst a small subset of the total respondent pool.

Data reliability was analysed by calculating internal consistency using Cronbach alpha tests separately with the PA and the NA scale items. To analyse the latent structure of the PANAS, a Confirmatory Factor Analysis (CFA) was performed as the PANAS has previously been validated with a theoretical structure of two correlated factors, which are repeatedly found within the model [32, 34]. There was a reasonable fit between the model and the observed data, and the detailed CFA model constructed is enclosed in the S2 File [35, 36].

Several statistical tests were conducted to assess the PA and NA scores differences between phases. Due to the large sample size, histogram and normal Q-Q plots were first used to assess the data distribution visually; the PA score generally fits a normal distribution, and the NA score is skewed (see S3 File). Therefore, for the PA scores, the one-way ANOVA was used to test the overall group difference among phases, and the Tukey-Kramer test was used to perform multiple pairwise comparisons between the means of the groups. For the NA scores, the Kruskal-Wallis rank-sum test was used to test the overall group difference among phases, and the Wilcoxon rank-sum test was used for performing multiple pairwise comparisons between the median of the groups. The null hypothesis is that there is no significant difference in the scores between the tested phases.

One-way ANOVAs were performed to analyse the influence of demographic factors, including age, gender, education, employment, number of households, and garden ownership, on the separate PA and NA overall scores. Using the ANCOVA model, the significant factors were selected as potential covariates in subsequent models on differences among the phases in the mean scores on the PANAS. Confidence intervals were based on 1000 bootstrap samples because of the skewness of the distribution of the negative scores. Post-hoc tests were used to examine the differences between the six phases.

A separate analysis was performed among the subset of repeated measures to assess the individual's adaption throughout the six phases. Paired Wilcoxon signed-rank tests were used to check whether there were significant differences in the PA and NA mean scores separately between the adjacent phases. For the sake of interpretation and visualisation of the interaction effect, we categorised the PANAS scores into five intervals (0–10, 10–20, 20–30, 30–40, 40–50). Then, alluvial plots were used to display the movement of scores between phases.

The data analysis and visualisation were performed using R Studio version 2021.09.1 [37]. An alpha level of 0.05 was established as the criterion for statistical significance for all analyses done (p-value < 0.05).

## Results

### Sample description

The total sample comprised 8073 records from 4222 participants; of them, 167 people were included as repeated measures. Within both samples, most participants are female and white with higher education backgrounds (bachelor's degree and post-graduate degree). It is worth noting that this over-representation of a particular demographic group may affect the

interpretation of the results, especially as existing evidence suggests that mental health was more adversely affected in demographic groups with pre-existing health inequalities, which may have been exacerbated during the pandemic [23, 24, 26]. About half of them were employed or self-employed. In terms of the spatial distribution, the bulk of the participants stayed in the south part of England (Top 3 regions: South East 22.52%, London 13.50%, and South West 10.59%). Table 2 presents the demographic details of the two samples. The participant details of each study phase were disclosed in the S1 Table.

In terms of the PANAS score, Table 2 shows the average PA and NA scores for each demographic group. Table 3 presents the results of the PANAS scores by study phase, allowing researchers to look at the emotional states of the study participants over time. As illustrated in the table, the PA scores in phase 2, phase 5, and phase 6 exceed the average, and the NA scores in phase 1, phase 2, phase 3, and phase 4 are above the average score. The one-way ANOVA test result for the PA scores shows a significant change throughout the phases. Furthermore, the Tukey-Kramer test result shows that the differences between phases 2 and 3, phases 3 and 4, and phases 4 and 5 were significant, with an adjusted p-value below 0.05. The Kruskal-Wallis rank-sum test results show a significant difference in the NA scores. Further, the Wilcoxon test result shows that the differences between phase 3 and phase 4, phase 4 and phase 5, and phase 5 and phase 6 are statistically significant with an adjusted p-value below 0.05. The

**Table 2. Demographics and overall PANAS scores on average.**

| Sample | Unique participants | Repeated measures | Overall PA Score | Overall NA scores |
|---|---|---|---|---|
| Sample size | 4222 | 167 | 8073 | |
| Measures | Count (percentage) | Count (percentage) | Mean (±SD) | |
| **Gender** | | | | |
| Female | 3521(83.40%) | 132(81.99%) | 27(±8.05) | 19.7(±7.3) |
| Male | 650(15.40%) | 24(14.91%) | 27.4(±7.95) | 18.3(±7.7) |
| **Age** | | | | |
| 18–24 | 120(2.84%) | 4(2.48%) | 23.6(±6.68) | 22.8(±8.35) |
| 25–34 | 278(6.58%) | 13(8.07%) | 25.9(±7.61) | 22.2(±8.05) |
| 35–44 | 409(9.69%) | 12(7.45%) | 25.2(±7.87) | 22.8(±7.77) |
| 45–54 | 770(18.24%) | 25(15.53%) | 26.9(±8.05) | 20.4(±7.67) |
| 55–64 | 1414(33.49%) | 48(29.81%) | 27(±7.99) | 19(±7.12) |
| 65+ | 1204(28.52%) | 56(34.78%) | 28.2(±8.07) | 17.5(±6.43) |
| **Employment** | | | | |
| Employed | 2328(55.14%) | 84(52.17%) | 26.8(±7.98) | 20.2(±7.49) |
| Not employed | 419(9.92%) | 13(8.07%) | 25.2(±8.07) | 21.8(±8.26) |
| Retired | 1383(32.76%) | 61(37.89%) | 27.9(±7.98) | 17.6(±6.54) |
| **Education** | | | | |
| Post-graduate degree | 1334(31.60%) | 57(35.4%) | 27.5(±8.02) | 20(±7.52) |
| College or university | 1977(46.84%) | 79(49.07%) | 27.2(±7.9) | 19(±7.14) |
| Higher or secondary or further education | 587(13.91%) | 18(11.18%) | 25.5(±8.04) | 19.9(±7.85) |
| Secondary up to 16 years | 298(7.06%) | 4(2.48%) | 25.9(±8.71) | 19.3(±7.64) |
| Less than secondary | 25(0.59%) | 0(0%) | 25.6(±9.93) | 22.2(±9.66) |
| **Household number** | | | | |
| Alone | 860(20.37%) | 26(16.15%) | 26.8(±8.14) | 18.9(±7.19) |
| 2 | 1970(46.66%) | 90(55.9%) | 27.2(±8.09) | 19.2(±7.41) |
| 3 | 635(15.04%) | 25(15.53%) | 26.5(±7.8) | 20.2(±7.7) |
| 4 | 516(12.22%) | 9(5.59%) | 27(±8.02) | 20.6(±7.39) |
| 5 or more | 216(5.12%) | 8(4.97%) | 27.5(±7.83) | 20.3(±7.08) |

**Table 3. PANAS scores overall in each phase and hypothesis test results between phases.**

| PANAS Score | | PA scores | | | | NA scores | | | | | | PA | | NA | |
|---|---|---|---|---|---|---|---|---|---|---|---|---|---|---|---|
| | N | Mean | SD | Min (10) | Max (50) | Mean | SD | Min (10) | Max (50) | | | F (df) | P-value | X² (df) | P-value |
| **Phase 1** | 2687 | 26.885 | 7.629 | 10 (0.37%) | 2(0.07%) | 20.118 | 7.301 | 103 (3.83%) | 0 | **Overall test**[a] | | 27.66 (5,8067) | <0.001 | 205.22 (5) | <0.001 |
| **Phase 2** | 343 | 27.344 | 8.529 | 4(1.17%) | 1(0.29%) | 19.781 | 7.581 | 14(4.08%) | 0 | **Post-Hoc test**[b] | | **Diff** | **P-value** | **Diff** | **P-value** |
| **Phase 3** | 1954 | 25.733 | 8.201 | 21(1.07%) | 3(0.15%) | 20.361 | 7.619 | 88(4.50%) | 1(0.05%) | **Phase 2–1** | | 0.459 | 0.917 | -0.337 | 0.206 |
| **Phase 4** | 1059 | 26.630 | 8.234 | 9(0.85%) | 3(0.28%) | 19.646 | 7.703 | 54(5.10%) | 1(0.09%) | **Phase 3–2** | | -1.611 | 0.007 | 0.58 | 0.159 |
| **Phase 5** | 1140 | 28.360 | 8.018 | 2(0.18%) | 2(0.18%) | 18.217 | 7.135 | 100 (8.77%) | 1(0.09%) | **Phase 4–3** | | 0.896 | 0.038 | -0.715 | 0.004 |
| **Phase 6** | 890 | 28.935 | 7.882 | 4(0.45%) | 2(0.22%) | 17.181 | 6.450 | 83(9.33%) | 0 | **Phase 5–4** | | 1.730 | 0.001 | -1.429 | 0.001 |
| **Overall** | 8073 | 27.027 | 8.038 | 50(0.62%) | 13 (0.16%) | 19.508 | 7.409 | 442 (5.48%) | 3(0.04%) | **Phase 6–5** | | 0.575 | 0.590 | -1.036 | 0.004 |

[a]: Overall test was done by using ANOVA for positive score and Kruskal-Wallis rank-sum test for the negative score.

[b]: Post-hoc test was done using Tukey-Kramer test (95% family-wise confidence level) for positive score and Wilcoxon rank-sum test for the negative score; only the results of two adjacent phases are recorded.

frequencies and percentages of minimum (10) and maximum (50) scores were tabulated for each sample to examine the potential floor and ceiling effects.

## Positive and negative emotions across key demographics

Fig 2 shows how the PANAS scores were associated with demographic variables. In general, gender, age and education significantly affected the PANAS scores. The ANOVA test result (see S2 Table) shows that males have higher PA scores and lower NA scores than females. People aged 18–24 have the lowest PA and higher NA scores, while older adults over 65 have the highest PA scores. People who are retired have the highest PA score, while unemployed people have the lowest PA and the highest NA scores. People who live in four households have the lowest NA scores (p<0.001), and people who live in two households have significantly higher PA scores. There is no clear pattern among the education groups.

## Identifying hard times: A cross-sectional look at positivity and negativity at points throughout the pandemic

A Chi-square test was used to determine whether these variables showed phase differences in their distribution, which turned out to be true for all the demographic variables (see S3 Table). These variables thus served as covariates in the subsequent model on differences between the phases in the mean scores on the PANAS (ANCOVA). Confidence intervals were based on 1000 bootstrap samples because of the skewness of the distribution of the negative affect scores. Post-hoc tests were used to examine the differences between the six phases.

Fig 3 shows the estimated mean scores per phase of the PANAS after adjustment for covariates. The overall test for the adjusted mean differences between the phases was significant for both PA and NA scores. There was a significant difference in mean PA [$F(5,7449) = 28.607$, p <0.001] between the six phases whilst adjusting the demographic covariates. Post-hoc tests for the PANAS showed that participants scored significantly higher on the PA score in phase 6 (Estimated Mean = 27.2; 95% CI: 26.4 to 28.1; p<0.001), phase 5 (Estimated Mean = 26.8; 95% CI: 26.0 to 27.6; p<0.05), and phase 2 (Estimated Mean = 26.1; 95% CI: 24.9 to 27.3; p <0.05), and significantly lower in the other three phases. Similarly, it also shows lower NA scores in

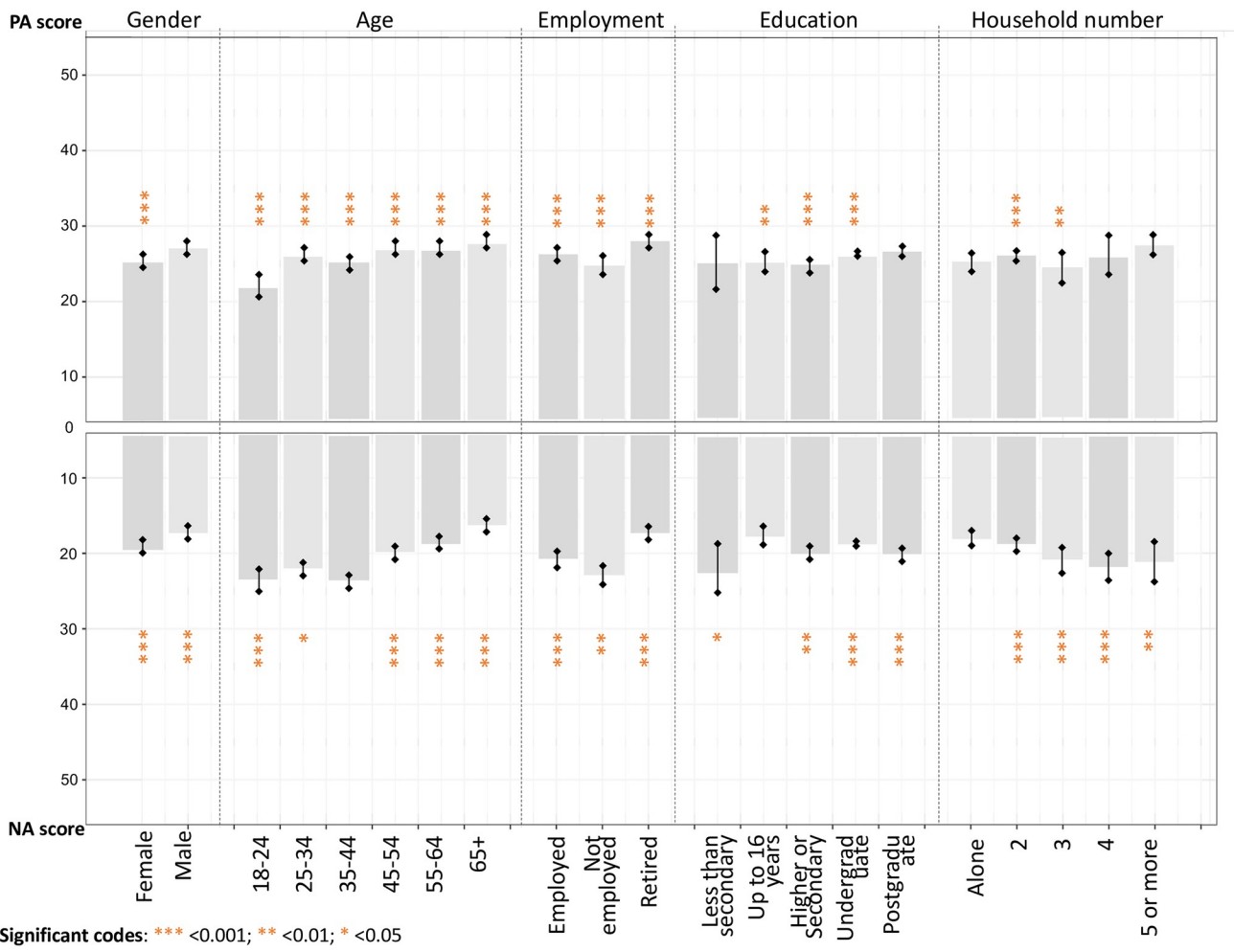

**Fig 2. Estimated demographic difference of the positive and negative scores among the whole samples.**

phase 2 (Estimated Mean = 20.8; 95% CI: 19.7 to 21.9; p <0.05), phase 5 (Estimated Mean = 20; 95% CI: 19.3 to 20.8; p <0.05) and phase 6 (Estimated Mean = 19.2; 95% CI: 18.4 to 20; p <0.05), which indicate that people tend to have better emotional status in these three phases.

### Emotional ups and downs: Following the cohort throughout the pandemic

The change in positive and negative effects among the repeated measures throughout the 5 phases was illustrated in the Alluvial diagrams (Fig 4). Each colour represents 10-point intervals from the lowest 0–10 in dark green to the highest 40–50 in light green. The higher scores represent more positive or more negative. The Wilcoxon signed-rank test was used to test the change between the adjacent phases. As indicated in the graph, there is no significant difference for both positive and negative scores between phases 1 vs. 3 and phases 3 vs. 4, with all p-values greater than 0.05. Nevertheless, from phase 4 to phase 5, the PA score significantly increased while the negative score decreased considerably, with both p-values less than 0.05, which indicates this population has more positive emotions in phase 5 than in phase 6.

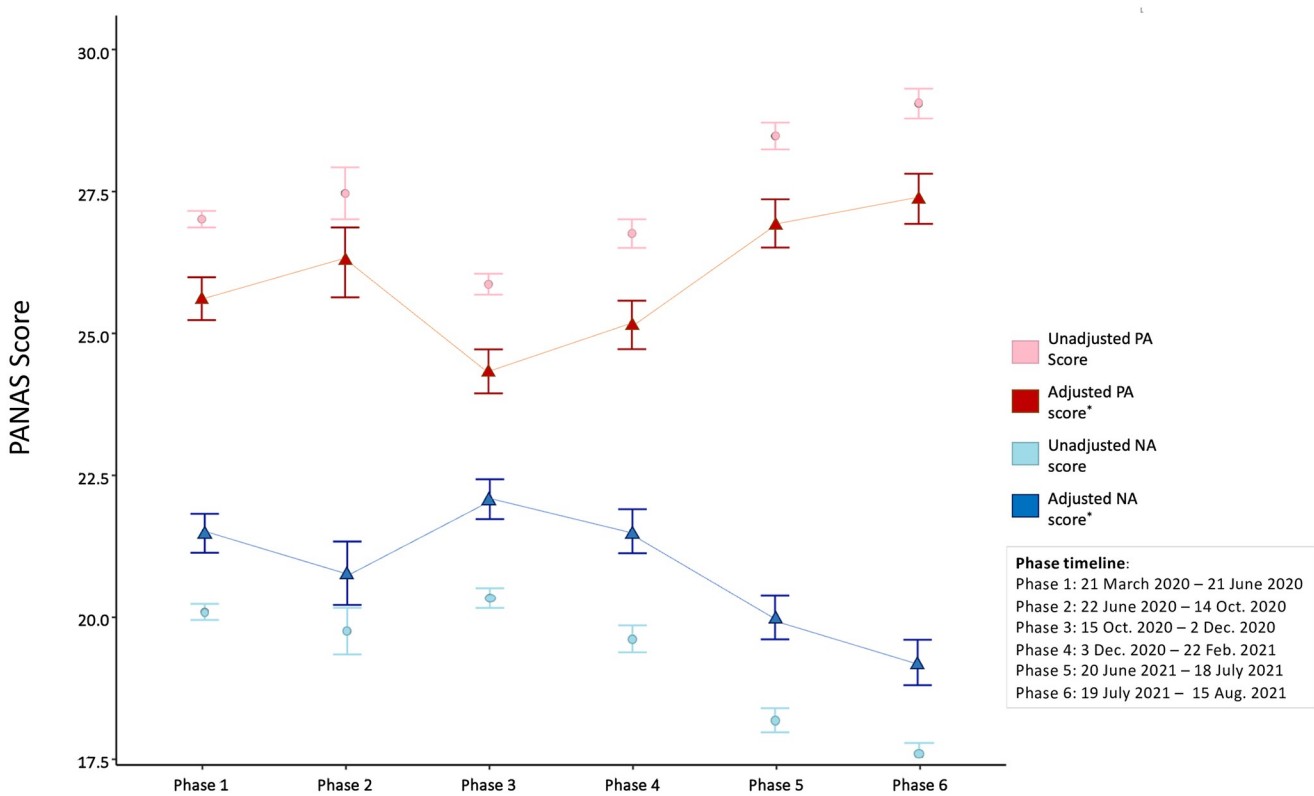

**Fig 3. The adjusted model results display the PANAS score difference among phases.** *Estimated means and standard errors (SE) based on 1,000 bootstrap samples from ANCOVA models adjusted for age, gender, education, employment, and household number.

Similarly, the negative score also significantly decreased from phase 5 to phase 6 (p-value<0.05), although the PA score does not show significant changes, indicating people were generally less negative in phase 6 than in phase 5.

## Discussion

We performed a longitudinal survey study to assess the emotional states of people living in the UK during the COVID-19 pandemic. Researchers looked at various demographic groups across six phases using the Positive and Negative Affect Schedule (PANAS) to understand key differences in emotional states across groups. A cross-sectional, as well as a cohort approach to data analysis was used to understand the general emotional state at certain moments during the pandemic and the change in emotional state through time. We summarise the key findings, the strengths and limitations of our study and provide recommendations for future research in this section.

### Key findings

Demographics that were found most significant to PANAS score were gender, age and employment. In general, our study found that gender had a significant relationship with reported emotional state. Throughout the study phases, females, on average, reported a higher PANAS score (more positive emotions) than male respondents. This is surprising in light of multiple findings that suggest that women were more likely to suffer more mental health

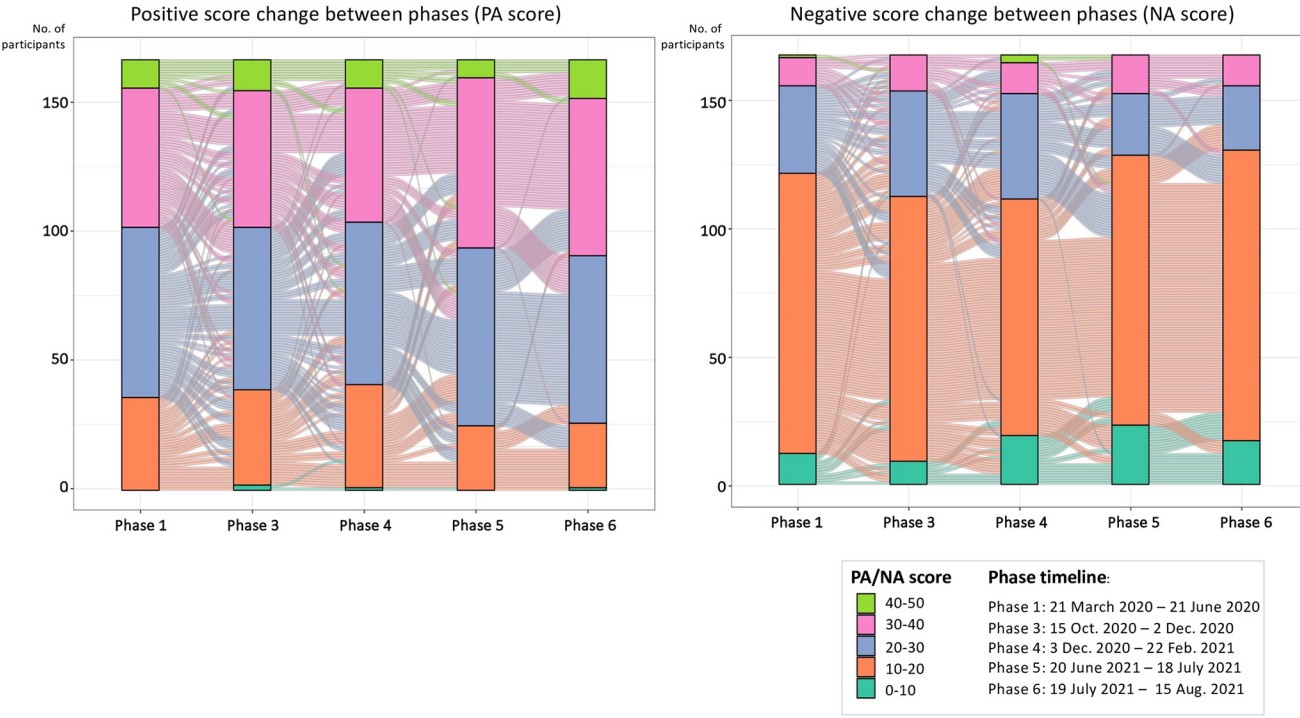

**Fig 4.** The dynamics of the PANAS scores change among phases shown by the alluvial plots (Left: PA score; Right: NA score; each line is represented by a line connecting their PANAS scores in each phase, coloured by the former phase) (N = 167; Phase 2 was excluded due to insufficient sample size). **Notes**: (1) Size of the bar represents the number of participants whose PANAS score falls into the corresponding range; (2) Coloured lines represent the PANAS scores of participants moving between phases, coloured by the former phase; (3) Phase 2 was excluded due to insufficient sample size.

problems during lockdown [18, 23, 24, 38, 39]. In addition, our study found that older adults (aged 55 and above) tended to exhibit more positive and less negative emotional responses compared to younger individuals, which is consistent with a large-scale study conducted in the UK [40]. It is found that younger individuals reported higher levels of anxiety, depression, and loneliness during lockdown, as well as an increase in loneliness over time, while older adults showed greater resilience to negative emotions [40]. Older females, which made up a large part of the sample, were generally very positive and less negative than younger groups. This differs from the findings presented by Ramiz et al., which demonstrated that older females were more likely to have symptoms of depression and anxiety during the lockdown [23]. These discrepancies may be explained by the following reasons: firstly, it is important to note that most previous studies focused on psychological disorders or symptoms, such as anxiety, depression and suicide. In contrast, this study explored the emotional status using the PANAS scale, which measures emotional affect rather than psychological symptoms. Measuring emotions involves capturing subjective, momentary experiences, which can be influenced by a wide range of situational and environmental factors, while psychological conditions are typically assessed using standardised diagnostic criteria that require a certain level of symptom severity and duration [41, 42]. As such, emotions can be more variable and dynamic compared to psychological conditions. Furthermore, this study takes the longer pandemic period into account, while other studies only focus on the initial stage of lockdowns. Therefore, the inconsistency may partly be explained by the different measurements and time frames. Further qualitative studies will be helpful in identifying the detailed reasons and placing these results into context.

Regarding employment, retired people reported more positive and less negative emotional states, while people who were not employed were the most negative. Employment status posed a significant challenge during the pandemic, with 24% of UK jobs at risk during the lockdowns [43]. Despite the age bracket that many retirees fall into, which is generally considered 'high-risk' in terms of health concerns during the pandemic, this demographic reported more positive and less negative feelings. This may be attributed to the fact that losing a job due to the pandemic is a large source of fear and anxiety [44–46]. This also could be attributed to the fact that many retirees' activities and lifestyles were less affected by lockdowns, as their commutes to work, occupation, and child-rearing were frequently unchanged [29, 47].

The above demographic factors were subsequently considered in constructing the ANCOVA model to balance the demographic difference between samples. The adjusted result shows people are more positive in phases 5 and 6 (which displays the higher adjusted PA score and lower adjusted NA score) and more negative in phases 3 and 4. The individual PANAS score change was finally analysed among the repeated measures. The result shows that people have higher PA scores and lower NA scores in phase 5 and phase 6, which are consistent with the adjusted model, indicating that people have more positive emotions in phase 5 and phase 6 after the so-called 'Freedom Day' on July 19, 2021. Phases 3 and 4 represent some of the most restrictive lockdowns in the UK. Phases 2 and 3 came after the first lockdown, which may show less negative reaction based on the novelty and hope inherent in the earliest days of the pandemic. While surely full of uncertainty, the first lockdown was also a time when people thought our collective engagement with COVID-19 may be a couple of weeks long. Unsurprisingly, phases 3 and 4 show a higher level of negative emotions such as sadness, fear, and anxiety. Phases 5 and 6 are the phases in which negative emotions are less strong, and positive emotions were reported at a higher level. Phases 5 and 6 include the general relaxing of COVID-19 restrictions, as well as the commencement of 'Freedom Day' in the UK, setting for the end of all COVID-19 restrictions. Unsurprisingly, there are generally more positive emotional states reported during this time.

It is worth considering that the emergence of community-based mutual aid groups and activities such as NHS volunteering and clapping for healthcare workers in the early stages of the pandemic provided a sense of solidarity and support for individuals during a time of uncertainty and fear [48–50]. However, as the pandemic continued and local services began to catch up with demand, people may have experienced a sense of decreased reliance on their communities and a shift towards negative emotions, such as anxiety and fear [51]. This may have contributed to the change in emotional responses observed between the early and later stages of the pandemic. A longitudinal study in the UK found a decline in physical activity during the early stage of the COVID-19 pandemic [40]. Given the established link between physical activity and the health [52–54], it may also partly explain the difference in emotional responses between phase 1 and phase 2. However, further work may be important to understand who may be reporting increased negative emotions during periods of relaxed restriction, including vulnerable groups such as the immunocompromised, who are protected by stringent mask-wearing or social distancing.

This study offers valuable insights into the emotional challenges and strengths individuals faced during the pandemic, providing essential information to shape policies and interventions that support emotional well-being in crisis situations. The empirical evidence presented in this research highlights the detrimental emotional impact of lockdowns and uncertainty during the early stages of the COVID-19 pandemic, particularly among specific demographic groups. Monitoring emotional changes becomes crucial in comprehending the complex experiences people undergo when public health measures are enforced, thereby informing future health policies for potential pandemics. The findings indicate that younger individuals, males,

and the unemployed were more prone to negative emotions during the pandemic. To address this, targeted interventions from government, local authorities, universities, and voluntary organisations might be necessary. Such interventions could be vital not only during pandemics but also in addressing the lasting effects on social networks and employment for these specific groups. Future research could focus on investigating the long-term emotional experiences and well-being of diverse populations affected by the COVID-19 pandemic. Additionally, examining the effectiveness of interventions aimed at promoting positive emotional responses would be beneficial in refining strategies for handling similar crises in the future.

### Strengths, limitations and further research

To our knowledge, the present study is the first to evaluate the emotional status using the PANAS in a British population throughout the pandemic, assessing the emotional impact of various stages of restrictions. Secondly, it has rich data, which included 4,222 unique participants and 6 phases at different time points of the pandemic, covering three main lockdowns and the early post-lockdown stage in the UK. The mixture of cross-sectional and prospective-cohort methods also provided robust evidence to support the conclusions.

However, there are several limitations to consider. Firstly, the study is not based on a nationally representative sample, although it does have a wide inclusion across all socio-demographic groups. The result is more representative of older and highly educated females living in the south part of England, which has more affluent and urban areas, particularly in the southeast region. While we have taken measures to ensure a diverse sample, we acknowledge that our findings may not generalise to other populations or individuals in regions with higher levels of social deprivations or large northern cities. It is possible that diverse experiences were not adequately captured. However, we sought to correct some of this bias by using statistical methods, although we acknowledge the limited generalizability and a note of caution in interpreting the findings. Secondly, we acknowledge that the reliance on convenience sampling through social media may have led to biased sampling and may not represent the wider population. By recruiting online, we may have missed out on reaching certain populations who do not engage with social media, such as older generations or those with limited access to technology. It is important to note that these limitations may affect the generalizability of our findings and should be considered when interpreting the results. Future research should consider more representative sampling methods to address these limitations. Thirdly, since the PANAS score is a self-reported scale, the length of the questionnaire may affect people's answers. However, we tested the data reliability using multiple measurements, and the possible contaminated data were removed for analysis. In addition, the measure did not capture what or why participants were scoring differently by phases. Thus, it was impossible to assess if other personal or social reasons contributed to the changes.

Because of the need for more representation within our sample, we are keen to analyse the data for women in the older age bracket as a stand-alone analysis. While this group could be considered 'high risk' for serious complications due to COVID-19, it was also some of the most positive in our study. An analysis of the emotional states and the activities of this demographic is vital for understanding resilience during a pandemic in future work. It is interesting to identify key protective factors to the positivity of this group during challenging times.

### Conclusion

Disasters are one of the most significant life stressors, with the potential to affect our mental well-being and even our ability to recover. Taking inventory of emotional response to disaster is a critical piece in understanding individual and community capacity for resilience. While

our findings suggest that older people, retirees, and women generally reported more positive moods throughout the pandemic, and people generally reported more positive feelings in the summer of 2021, it is important to note that this study primarily assessed a group of well-educated older women living in a relatively affluent region. As such, our conclusions should be considered exploratory in nature and not generalisable to all older adults in the UK. Nevertheless, our study highlights the importance of collecting data on positive mood states during pandemics to inform policy and communication strategies. Going forward, future studies should aim to include more diverse samples, collecting data on both positive and negative mood states, as well as symptoms of psychological distress, during pandemics to provide a more holistic picture that can inform policy and communication strategies. Despite the potential selection bias and limitations, our study offers valuable insights into the emotional ups and downs experienced during a pandemic and can help identify particularly challenging moments, as well as groups that exhibit particular resilience and hope.

## Supporting information

**S1 File. Questionnaire.**
(DOCX)

**S2 File. CFA analysis result for assessing internal reliability.**
(DOCX)

**S3 File. Histogram and Q-Q plot for checking data distribution.**
(PDF)

**S1 Table. Participants' socio-demographic distribution.**
(XLSX)

**S2 Table. ANOVA test for each demographic subgroup result.**
(XLSX)

**S3 Table. Chi-square test for assessing demographic differences among phases.**
(XLSX)

## Author Contributions

**Conceptualization:** Lan Li, Caroline E. Wood, Philip Baker, Patty Kostkova.

**Data curation:** Lan Li, Anwar Musah, Philip Baker.

**Formal analysis:** Lan Li.

**Funding acquisition:** Patty Kostkova.

**Investigation:** Lan Li, Philip Baker.

**Methodology:** Lan Li, Ava Sullivan, Anwar Musah, Katerina Stavrianaki, Caroline E. Wood, Patty Kostkova.

**Project administration:** Patty Kostkova.

**Resources:** Patty Kostkova.

**Software:** Lan Li.

**Supervision:** Patty Kostkova.

**Validation:** Patty Kostkova.

**Visualization:** Lan Li.

**Writing – original draft:** Lan Li, Ava Sullivan.

**Writing – review & editing:** Lan Li, Ava Sullivan, Anwar Musah, Katerina Stavrianaki, Patty Kostkova.

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
