## [Decision Letter · Decision Letter 0]

2 Sep 2022

PONE-D-22-14167Positive and Negative emotions during the COVID-19 Pandemic: A longitudinal survey study of the UK Population

PLOS ONE

Dear Dr. Li,

Thank you for submitting your manuscript to PLOS ONE. The paper has been read by two competent reviewers. After careful consideration of the reports and my own reading of the paper, we feel that it has merit but does not fully meet PLOS ONE’s publication criteria as it currently stands. Therefore, we invite you to submit a revised version of the manuscript that addresses the points raised during the review process.

In addition to accounting for each referee comment, please address the following points:

- There is a sizable literature on mental health (MH) during COVID times which present causal evidence on exposure to pandemic induced lockdown and mental heath exploiting spatial variation in lockdown timing. The manuscript needs to acknowledge this - please update your literature review accordingly - and offer some discussion of how your approach compares and contrasts with other key studies on the topic. 

- Following from the above comment and taking note of concerns of selection  bias highlighted by Referee-1, I expect a more carefully written version in the next round. Since the manuscript claims to offer "creative exploratory analysis and statistical tests" and in conclusion "we performed a longitudinal survey study", please clarify how you achieved this from a methodological point of view that is consistent with the literature  (including a discussion of selection  bias issues) - else, add more to rigor to your empirical analysis during the revision. I say this b/c the dataset is very rich - PANAS longitudinal survey from the UK (March 2020 to July 2021). Yet the analysis is only compares MH (+ve and -ve emotions) across different phases of lockdown. As such, the dynamic nature of the data is not fully exploited. 

- Figures 4 and 5 are not acceptable in terms of quality and format. Please use "actual dates" or "text label" to clearly identify each phase on the x-axis instead of numbers 1-7 (you already have the name description of each phase in Fig 1). Otherwise it's difficult for readers to remember which number corresponds to which phase.

- In Figs 4 & 5, the legends used are not standard and not aligned with terms used in the main text. What do you mean by "Original positive score"? Either say "unadjusted score (positive)" or "raw score (positive)" 

- Fig 2 looks strange. Consider adding corresponding mean values etc and add notes at the bottom to explain what the content stands for. The title is also very confusing "Positive and negative affect states measured by PANAS" b/c there is nothing on "measurement scale" in Fig 2. Please address similar concerns for all other Tables and Figs.

We look forward to receiving your revised manuscript.

Kind regards,

M Niaz Asadullah

Academic Editor

PLOS ONE

Journal Requirements:

Reviewers' comments:

Reviewer's Responses to Questions

**Comments to the Author**

1. Is the manuscript technically sound, and do the data support the conclusions?

Reviewer #1: Partly

Reviewer #2: Yes

2. Has the statistical analysis been performed appropriately and rigorously? 

Reviewer #1: No

Reviewer #2: Yes

3. Have the authors made all data underlying the findings in their manuscript fully available?

Reviewer #1: No

Reviewer #2: Yes

4. Is the manuscript presented in an intelligible fashion and written in standard English?

Reviewer #1: Yes

Reviewer #2: Yes

5. Review Comments to the Author

Reviewer #1: Comments on “Positive and Negative emotions during the COVID-19 Pandemic: A longitudinal survey study of the UK Population”

This is an interesting study taking advantage of a panel of six rounds where a sample of respondents participated in all six rounds. Such balance panels are typically hard to find, and it is commendable that the authors could take advantage of this aspect of the survey. My main concern is the sample size and nature of selection. Both of these issues will have implications for how generalizable the findings are at the national or “societal” level.

1. There are two types of selection issues here. One is the recruitment of the respondents was done through social media advertisement along with few other channels (p.6, lines 99-102). Hence, the people who responded may not be representative of the adults at the national level. While the time trend for the respondents in the balanced panel can be useful, any inference drawn using 167 respondents worries me. Few suggestions:

(a) How about reporting the average for the entire sample by rounds? Didn’t they take same PANAS tests? It seems so from Table 1. So, the summary stats based on repeated cross-sectional data will also be very useful as they may have more statistical information.

Another benefit of showing the results from the balanced panel (N = 167) and overall sample (which varied by rounds) is that it will show the robustness of the findings and allow how representative the restricted sample is.

(b) Should we consider deriving some sample weights using population wide data? Obviously, people who self-selected to take part in the survey and further to participate in all six rounds may not be similar to the population. But sample weights can perhaps partially correct the biases.

2. The authors have been able to show that the changes in moods (positive and negative) coincide with the mobility restrictions. Authors may want to present the trends superimposed on a chart which shows the degree of restrictions over time. This will be a very useful visual aide to understand the association between COVID restrictions and mood of people sampled.

3. The authors claim about sample size to be large (p.16, lines 303-304). Should a sample size of 167 be considered large?

4. In terms of analyses, the authors have some covariates which are systematically correlated with the PANAS score. Authors may consider taking different respondent features (gender, age, etc.) and interact with round or the strictness of lockdown. The coefficients for the interaction terms may allow testing how different features are associated with the round specific mood outcomes.

5. Associations of moods with some of the features need some elaboration. Why did women report better mental health outcomes compared to men? Is this typical in the literature? In general, the authors to discuss their findings in the context of what other studies have found so far.

Reviewer #2: The manuscript is well written except a couple of typos scattered throughout text. The paper uses PANAS emotional state measures of positive and emotional states of 4222 people (and 167 observed repeatedly) in UK between March 2020 and October 2021. Two separate study designs are used. The first one is the cross-sectional study which looks at the difference of the emotional states across various demographic groups. The second one is a prospective cohort study that explores the changes throughout the stages of the pandemic.

All the tables for the analysis are included, though some need more sharpness.

6. PLOS authors have the option to publish the peer review history of their article (what does this mean?). If published, this will include your full peer review and any attached files.

Reviewer #1: No

Reviewer #2: No

---

## [Author Response · Author response to Decision Letter 0]

7 Oct 2022

Thank you for giving us the comments. Please find the file entitled "Response to Reviewer" attached for details. Many thanks!

---

## [Decision Letter · Decision Letter 1]

15 Mar 2023

PONE-D-22-14167R1Positive and Negative emotions during the COVID-19 Pandemic: A longitudinal survey study of the UK PopulationPLOS ONE

Dear Dr. Li,

Thank you for submitting your manuscript to PLOS ONE. After careful consideration, we feel that it has merit but does not fully meet PLOS ONE’s publication criteria as it currently stands. Therefore, we invite you to submit a revised version of the manuscript that addresses the points raised during the review process.

We look forward to receiving your revised manuscript.

Kind regards,

Guglielmo Campus, Ph.D DDS

Academic Editor

PLOS ONE

Reviewers' comments:

Reviewer's Responses to Questions

**Comments to the Author**

1. If the authors have adequately addressed your comments raised in a previous round of review and you feel that this manuscript is now acceptable for publication, you may indicate that here to bypass the “Comments to the Author” section, enter your conflict of interest statement in the “Confidential to Editor” section, and submit your "Accept" recommendation.

Reviewer #3: (No Response)

Reviewer #4: (No Response)

2. Is the manuscript technically sound, and do the data support the conclusions?

Reviewer #3: Yes

Reviewer #4: Partly

3. Has the statistical analysis been performed appropriately and rigorously? 

Reviewer #3: Yes

Reviewer #4: I Don't Know

4. Have the authors made all data underlying the findings in their manuscript fully available?

Reviewer #3: Yes

Reviewer #4: No

5. Is the manuscript presented in an intelligible fashion and written in standard English?

Reviewer #3: Yes

Reviewer #4: Yes

6. Review Comments to the Author

Reviewer #3: Thank you for your manuscript. This is the first time I have seen the paper and have not been able to see your previous response to reviewers, hence my comments take the manuscript without prior knowledge of reviews and on face value.

You present a clear introduction and rationale as to why this study may be needed considering there is limited data assessing changes in the pandemic and individuals responses to it over time. My initial impression is that the discussion could be expanded to go into more depth and engage with more literature. There are some key limitations not mentioned that should be added. Lastly I believe there should be more concrete implications added to the manuscript to answer the 'so what' question, i.e. describing why your findings are useful and what policies, practice or research it can inform in the future. I add more detail on these issues below. I believe if these are addressed, the work will provide an interesting piece of evidence in the COVID literature in the UK.

Line 94 – typo – should this be 'longitudinally'?

Line 279 – why might measuring emotions, rather than psychological conditions have different presentation in the data? You state this may be the reason for finding difference, but then do not explain why this might cause a difference - e.g. I assume it is because emotions can flux more readily that probable conditions - but citing supporting research and explaining further would be useful.

- Further discussion of more context of phases would be useful e.g. In the first phases there were community responses in support and action – neighbourhood and NHS volunteering, clapping emergency support workers (aka we are all in this together - however some of this community spirit may have dissipated at later restricted phases?

- There is a huge amount of evidence collected overtime from the UCL Social Study, but this does not seem to be included or discussed in the discussion - it might benefit from including some of their findings in the context of your study. Additionally what international studies an you cite that have explored longitudinal data and how does this relate to your findings?

Line 321 –you state the geographical limitation but don't explain how this might bias results i.e. may not take into account other areas with higher levels of social deprivation, large northern cities etc. but why limitation –

Line 343 – do you mean 'cross-sectional' ?

Limitations –you make no mention of method of recruitment as being a limitation and possible source of bias – e.g. convenience sampling through social media online. It would be good to add how this might bias who took part in your study. e.g. possible that those who had experienced more problems would be more attracted to take part - it is known in other psychological studies that those with mental health problems are more likely to take part in studies that brand themselves as mental health studies and hence may bias average distress reported in those studies. Additionally by recruiting online and through social media, may bias which populations engage with facebook (for example) and you may not have reached younger or older generations - some noting or discussion of this would be useful.

Implications - It would be useful to discuss what implications these findings have and how/who they might be useful for - e.g. government policy? healthcare practice? future research. Currently the 'so what' question isn't answered fully enough to make these results feel tangible - hence some more discussion of tangible implications would be good.

Reviewer #4: This is manuscript presents an interesting exploratory analysis of changes in positive affect in a longitudinal cohort. The sample available for analysis is relatively small (n=167) and selection bias is a limitation to this study. However, the work does highlight the need to investigate changes in positive as well and negative affect in crisis situations as both dimensions are important to well-being. The PANAS also taps into fluctuation in mood states rather than assessing symptoms of psychological distress. As such, the study raises important questions for consideration and further investigation in developing pandemic response plans and policies.

I have read the previous reviewers' comments and author responses and can see that the manuscript has been significantly improved since the last version. However, there are still some issues that need to be addressed:

Abstract

• Consider rephrasing ‘paralysed the world’

• Ensure it is clear that it is positive affect sub-scale scores that are referred to in the results in this section, with higher scores indicating more positive affect.

Methods

• Page 6, lines 107-112. The sampling approach is a little unusual, with a second wave of recruitment taking place in October and December 2020. As such, these participants would not have completed the earlier surveys and would not have the full five data points available. There is also information about optional lifestyle questionnaires. Was the PANAS part of the optional questionnaire set or the core survey? There is some further information on selection of the sample in the ‘data cleaning’ section. This would be clearer and easier to follow in a single section, with a table or flow diagram to show exactly what data is available at each time point for cross-sectional analysis, and from all time points for longitudinal analysis.

• Analysis and results section mention the NA scale, but this is not covered in the abstract, introduction, or discussion.

Results

• P10, Line 187, as noted in the methods it is unclear what the 8073 records relate to – is this complete PANAS data available for analysis, and for how many unique individuals at each time point – or were these pooled for analysis?

• The demographic profile of the cohort indicates that well-educated middle and older-age females are considerably over-represented in the sample. This sampling issues are common with online surveys, but it is worth considering this during interpretation – particularly as there is ample evidence that mental health was more adversely affected in demographic groups where there were pre-existing health equalities which were exacerbated during the pandemic.

• Table 2 of the results does indicate numbers of data points available for analysis at each time point, and as noted above it would be good to see this come in earlier in the paper so that this is really clear.

• Given the extent of change in participants recruited and exiting the study at different time points, it would be useful to assess the potential for selective drop-out in terms of mental health – especially given the apparent improvement in positive affect during later stages in the study. I.e. were people with higher initial PA and lower NA more likely to be lost to follow-up?

Discussion

There are some interesting elements to the discussion, particularly challenging assumptions about older age and also highlighting the importance of employment and the impact of of of income for working age adults.

However, in this study, you are primarily assessing PA in a group of well-educated older women living in a relatively affluent region. There will be many older adults in the UK affected by poverty, ill-health, lack of digital literacy, social isolation, bereavement and so on who will have had quite different experiences over the course of the pandemic. Therefore, it is important not to over-generalise or overstate the conclusions. It needs to be clear that this is an exploratory analysis that raised interesting questions, but which cannot provide definitive answers at this stage. I would suggest pitching the conclusions more in terms of the importance of collecting data on positive mood states (as well as negative moods states and/or symptoms of psychological distress) during pandemics to enable a more holistic picture to inform policy and communication strategies.

7. PLOS authors have the option to publish the peer review history of their article (what does this mean?). If published, this will include your full peer review and any attached files.

Reviewer #3: No

Reviewer #4: No

---

## [Author Response · Author response to Decision Letter 1]

27 Apr 2023

Dear Reviewers,

Thank you for giving us the opportunity to submit a revised version of our manuscript titled ‘Positive and Negative emotions during the COVID-19 Pandemic: A longitudinal survey study of the UK Population’ to PLOS ONE to be considered for publication as Research Article.

We would like to express our gratitude for your time and effort in providing valuable feedback on our manuscript. We have carefully considered all comments and have made significant revisions based on the suggestions provided. We believe that the manuscript has improved substantially as a result of your feedback.

In order to make it easier to follow the changes we have made, we have included the comments provided by the reviewers in tables below, along with our responses and revisions. Please note that all page and line numbers cited in this document correspond to the file "Manuscript".

Once again, we thank you for your valuable feedback and hope that the revised manuscript meets your expectations.

We are very much looking forward to your decision and greatly appreciate your attention and valuable time.

---

## [Decision Letter · Decision Letter 2]

17 Jul 2023

PONE-D-22-14167R2Positive and Negative emotions during the COVID-19 Pandemic: A longitudinal survey study of the UK PopulationPLOS ONE

Dear Dr. Li,

Thank you for submitting your manuscript to PLOS ONE. After careful consideration, we feel that it has merit but does not fully meet PLOS ONE’s publication criteria as it currently stands. Therefore, we invite you to submit a revised version of the manuscript that addresses the points raised during the review process.

We look forward to receiving your revised manuscript.

Kind regards,

Marcus Tolentino Silva

Academic Editor

PLOS ONE

Journal Requirements:

**Additional Editor Comments:**

Introduction: (1) Clearly state the rationale for investigating emotions during the COVID-19 pandemic and link it to the psychological underpinnings. (2) Provide references to literature supporting the importance of emotions in predicting wellbeing and future mental health conditions.

Discussion: (3) Ensure that the added section in the discussion aligns with the findings presented in the study. (4) Clearly delineate the implications of the work, addressing why the study is interesting, important, and how it can inform policy and practice. (5) Avoid repetition between the first and last paragraph of the discussion. The first paragraph should focus on summarizing what was done and found, while the last paragraph should provide key "take-home messages."

Proofreading: (6) Thoroughly review the manuscript for typographical and grammatical errors. Conduct a comprehensive proofreading to ensure clarity and coherence of the writing.

Reviewers' comments:

Reviewer's Responses to Questions

**Comments to the Author**

1. If the authors have adequately addressed your comments raised in a previous round of review and you feel that this manuscript is now acceptable for publication, you may indicate that here to bypass the “Comments to the Author” section, enter your conflict of interest statement in the “Confidential to Editor” section, and submit your "Accept" recommendation.

Reviewer #3: (No Response)

Reviewer #4: (No Response)

2. Is the manuscript technically sound, and do the data support the conclusions?

Reviewer #3: Yes

Reviewer #4: Yes

3. Has the statistical analysis been performed appropriately and rigorously? 

Reviewer #3: Yes

Reviewer #4: Yes

4. Have the authors made all data underlying the findings in their manuscript fully available?

Reviewer #3: Yes

Reviewer #4: No

5. Is the manuscript presented in an intelligible fashion and written in standard English?

Reviewer #3: Yes

Reviewer #4: Yes

6. Review Comments to the Author

Reviewer #3: Thank you for your revisions and comments back to reviewers - the additions do improve the manuscript substantially.

Before this piece can be published - there is a key issue that I do not believe has been adequately addressed and this directly relates to the rationale for the study and what use understanding emotions has for healthcare professionals or policy makers in the future. In your introduction you state the aim of the work: 'This study aims to assess the change in emotions and feelings during the COVID-19 pandemic to the restrictions that were in place during the number of phases of the pandemics throughout the study period.' But you do not then go on to justify a rationale as to why you are investigating this? You state that others haven't measured emotions over a longitudinal period - but you do not address why investigating emotions is important. i.e. I assume investigating emotions is important because of links to wellbeing, predicting distress or future mental health conditions, important to know which groups are at risk and who may need support in the future. However you don't present a rationale or link to literature explaining why investigating emotions is important but make an assumption that emotions and wellbeing are tied - please cite some literature so readers can understand the psychological underpinning of where emotions sit in relation to predicting wellbeing or future mental health. This will then help your discussion.

In your discussion you have added a section from line 365 trying to address why this research is important and what implications it might have for policy and practice - yet it doesn't seem to entirely tie in with your findings - your findings demonstrate that those who were male, younger and not employed were more likely to have negative emotions over the pandemic - hence government, local authorities, universities and voluntary organisations may also need to look at how interventions can be targeted to these groups, not only during pandemics, but these groups may have lasting effects of the pandemic on their social networks and employment.

Thank you again - I believe the manuscript is nearly there, but I still think the rationale for the study and the implications are not clear enough yet.

Reviewer #4: The manuscript is much improved since the last version, with greater clarity around the exploratory nature of the study and what can and cannot be concluded from the findings.

There is significant repetition between the first and last paragraph of the discussion and this should be reviewed. The first paragraph should focus on a brief summary of what was done and what was found. The conclusions on summarising key 'take home messages'.

The writing is clear and of a high standard overall, but there are numerous typographical and grammatical errors throughout and thorough proof reading is required.

7. PLOS authors have the option to publish the peer review history of their article (what does this mean?). If published, this will include your full peer review and any attached files.

Reviewer #3: No

Reviewer #4: No

---

## [Author Response · Author response to Decision Letter 2]

10 Aug 2023

Dear Reviewers,

Thank you for giving us the opportunity to submit a revised version of our manuscript titled ‘Positive and Negative Emotions during the COVID-19 Pandemic: A Longitudinal Survey Study of the UK Population’ to PLOS ONE to be considered for publication as Research Article.

We would like to extend our sincere gratitude for your dedicated time and effort in reviewing our manuscript. Your valuable feedback has been immensely helpful in improving the quality of our work. We have taken all your comments into careful consideration and made revisions accordingly.

Notably, we have revised the rationale, strengthening its clarity and relevance. The discussion section has been thoroughly reviewed to ensure a more coherent presentation of our findings and a clear delineation of the implications of our work. Moreover, we have meticulously addressed any repetition between the discussion and conclusion, ensuring each section serves its distinct purpose.

To facilitate an easy understanding of the revisions, we have included the reviewer and editor's comments in the tables below, along with our responses and corresponding changes. All page and line numbers cited in this document correspond to the file "Manuscript."

We genuinely appreciate the time and expertise you have devoted to reviewing our work, and we are confident that the revised manuscript now meets your expectations.

Thank you once again for your invaluable feedback. We eagerly await your decision and remain grateful for your attention and valuable time.

---

## [Decision Letter · Decision Letter 3]

2 Jan 2024

Positive and Negative emotions during the COVID-19 Pandemic: A longitudinal survey study of the UK Population

PONE-D-22-14167R3

Dear Dr. Li,

We’re pleased to inform you that your manuscript has been judged scientifically suitable for publication and will be formally accepted for publication once it meets all outstanding technical requirements.

Kind regards,

Marcus Tolentino Silva, Ph.D.

Academic Editor

PLOS ONE

Additional Editor Comments (optional):

Reviewers' comments:

Reviewer's Responses to Questions

**Comments to the Author**

1. If the authors have adequately addressed your comments raised in a previous round of review and you feel that this manuscript is now acceptable for publication, you may indicate that here to bypass the “Comments to the Author” section, enter your conflict of interest statement in the “Confidential to Editor” section, and submit your "Accept" recommendation.

Reviewer #4: All comments have been addressed

2. Is the manuscript technically sound, and do the data support the conclusions?

Reviewer #4: Yes

3. Has the statistical analysis been performed appropriately and rigorously? 

Reviewer #4: Yes

4. Have the authors made all data underlying the findings in their manuscript fully available?

Reviewer #4: No

5. Is the manuscript presented in an intelligible fashion and written in standard English?

Reviewer #4: Yes

6. Review Comments to the Author

Reviewer #4: All recommended changes have been completed. The manuscript is now clear and provides an important insight into positive mood changes, as well as negative mood, during the COVID-19 pandemic, highlighting the need to make a more holistic assessment of well-being when evaluating the impact of pandemics and associated policies.

7. PLOS authors have the option to publish the peer review history of their article (what does this mean?). If published, this will include your full peer review and any attached files.

Reviewer #4: No

---

## [Editor Report · Acceptance letter]

29 Jan 2024

PONE-D-22-14167R3 

PLOS ONE

Dear Dr. Li, 

I'm pleased to inform you that your manuscript has been deemed suitable for publication in PLOS ONE. Congratulations! Your manuscript is now being handed over to our production team.

Kind regards, 

on behalf of

Prof Marcus Tolentino Silva 

Academic Editor

PLOS ONE